# Retinal Imaging-Based Oculomics: Artificial Intelligence as a Tool in the Diagnosis of Cardiovascular and Metabolic Diseases

**DOI:** 10.3390/biomedicines12092150

**Published:** 2024-09-23

**Authors:** Laura Andreea Ghenciu, Mirabela Dima, Emil Robert Stoicescu, Roxana Iacob, Casiana Boru, Ovidiu Alin Hațegan

**Affiliations:** 1Department of Functional Sciences, ‘Victor Babes’ University of Medicine and Pharmacy Timisoara, Eftimie Murgu Square No. 2, 300041 Timisoara, Romania; bolintineanu.laura@umft.ro; 2Center for Translational Research and Systems Medicine, ‘Victor Babes’ University of Medicine and Pharmacy Timisoara, Eftimie Murgu Square No. 2, 300041 Timisoara, Romania; 3Department of Neonatology, ‘Victor Babes’ University of Medicine and Pharmacy Timisoara, Eftimie Murgu Square No. 2, 300041 Timisoara, Romania; 4Field of Applied Engineering Sciences, Specialization Statistical Methods and Techniques in Health and Clinical Research, Faculty of Mechanics, ‘Politehnica’ University Timisoara, Mihai Viteazul Boulevard No. 1, 300222 Timisoara, Romania; stoicescu.emil@umft.ro (E.R.S.); roxana.iacob@umft.ro (R.I.); 5Department of Radiology and Medical Imaging, ‘Victor Babes’ University of Medicine and Pharmacy Timisoara, Eftimie Murgu Square No. 2, 300041 Timisoara, Romania; 6Research Center for Pharmaco-Toxicological Evaluations, ‘Victor Babes’ University of Medicine and Pharmacy Timisoara, Eftimie Murgu Square No. 2, 300041 Timisoara, Romania; 7Doctoral School, “Victor Babes” University of Medicine and Pharmacy Timisoara, Eftimie Murgu Square 2, 300041 Timisoara, Romania; 8Department of Anatomy and Embriology, ‘Victor Babes’ University of Medicine and Pharmacy Timisoara, 300041 Timisoara, Romania; 9Discipline of Anatomy and Embriology, Medicine Faculty, “Vasile Goldis” Western University of Arad, Revolution Boulevard 94, 310025 Arad, Romania; boru.casiana@uvvg.ro (C.B.); hategan.ovidiu@uvvg.ro (O.A.H.)

**Keywords:** oculomics, retinal microcirculation, cardiovascular disease, diabetes mellitus, deep learning, artificial intelligence, OCT, OCTA, fundus imaging

## Abstract

Cardiovascular diseases (CVDs) are a major cause of mortality globally, emphasizing the need for early detection and effective risk assessment to improve patient outcomes. Advances in oculomics, which utilize the relationship between retinal microvascular changes and systemic vascular health, offer a promising non-invasive approach to assessing CVD risk. Retinal fundus imaging and optical coherence tomography/angiography (OCT/OCTA) provides critical information for early diagnosis, with retinal vascular parameters such as vessel caliber, tortuosity, and branching patterns identified as key biomarkers. Given the large volume of data generated during routine eye exams, there is a growing need for automated tools to aid in diagnosis and risk prediction. The study demonstrates that AI-driven analysis of retinal images can accurately predict cardiovascular risk factors, cardiovascular events, and metabolic diseases, surpassing traditional diagnostic methods in some cases. These models achieved area under the curve (AUC) values ranging from 0.71 to 0.87, sensitivity between 71% and 89%, and specificity between 40% and 70%, surpassing traditional diagnostic methods in some cases. This approach highlights the potential of retinal imaging as a key component in personalized medicine, enabling more precise risk assessment and earlier intervention. It not only aids in detecting vascular abnormalities that may precede cardiovascular events but also offers a scalable, non-invasive, and cost-effective solution for widespread screening. However, the article also emphasizes the need for further research to standardize imaging protocols and validate the clinical utility of these biomarkers across different populations. By integrating oculomics into routine clinical practice, healthcare providers could significantly enhance early detection and management of systemic diseases, ultimately improving patient outcomes. Fundus image analysis thus represents a valuable tool in the future of precision medicine and cardiovascular health management.

## 1. Introduction

Artificial intelligence (AI) is revolutionizing medicine and diagnostics by enabling more accurate, efficient, and personalized healthcare. AI algorithms, particularly machine learning (ML) and deep learning (DL), can analyze vast amounts of data to identify patterns and make predictions that might be missed by human clinicians [1]. In conjunction with retinal imaging, AI can enhance diagnostic accuracy for various ocular and systemic conditions [2]. Advanced imaging techniques, such as Optical Coherence Tomography (OCT), provide detailed images of the retina, which AI algorithms can analyze to detect early signs of diseases like cardiovascular disease, diabetes mellitus complicated with retinopathy, neurodegenerative diseases, age-related macular degeneration, and glaucoma. AI-driven analysis of retinal images can help in early diagnosis and monitoring of these conditions, facilitating intervention and improved patient outcomes [2,3]. Studies have shown that AI systems can achieve diagnostic performance comparable to that of expert ophthalmologists, thus supporting clinical decision-making and reducing the burden on healthcare systems [1].

Oculomics leverages advanced imaging technologies, genetic information, and data analytics to understand systemic diseases and conditions through ocular biomarkers. The term “oculomics” originates from the combination of “oculo-”, referring to the eye, and “-omics”, which denotes a comprehensive study of a set of biological molecules. This term was first used in a scientific context in a paper published in 2020 [4]. By analyzing the retina, researchers can detect signs of systemic diseases such as diabetes, cardiovascular diseases, chronic inflammatory diseases and neurodegenerative conditions. The rich vascular structure and neural connections within the eye make it an excellent proxy for detecting early signs of these conditions [2,4].

Cardiovascular diseases (CVDs) are a leading cause of mortality globally, responsible for approximately 17.9 million deaths each year, which accounts for about 32% of all global deaths. The burden of CVD is particularly high in low- and middle-income countries, where over three-quarters of CVD deaths occur. Early detection and management of CVDs are crucial for reducing the associated morbidity and mortality [5]. The retina’s microvascular changes can reflect systemic vascular health, allowing for the detection of cardiovascular risk factors like hypertension and diabetes through non-invasive retinal scans [4]. Globally, the prevalence of diabetes has been rising rapidly. As of 2021, approximately 537 million adults aged 20–79 years are living with diabetes, and this number is projected to increase to 783 million by 2045. Early screening for diabetes is crucial for preventing the severe complications associated with the disease. Diabetes is a major cause of blindness, kidney failure, heart attack, stroke, and lower limb amputation. Early screening and detection are essential for managing the disease effectively and reducing its impact on individuals and healthcare systems [6].

Lastly, not only is it relatively straightforward to collect retinal images of a significant number of people, but it is also possible that this new field of study will make use of existing databases like MESA (Multi-ethnic Study of Atherosclerosis) [7], ACCORD (Action to Control Cardiovascular Risk in Diabetes) [8], and ARIC (Atherosclerosis Risk in Communities Study) [9]. Advances in the use of AI and ML approaches to retinal image processing can significantly enhance clinical practice in assessing, predicting, and managing hypertension, CVD, and peripheral arterial disease (PAD).

The aim of this review is to comprehensively examine the emerging field of oculomics, focusing on its potential to revolutionize the diagnosis and management of systemic diseases through the analysis of ocular biomarkers. By integrating advanced imaging technologies, genetic data, and artificial intelligence, oculomics aims to provide a non-invasive, cost-effective, and efficient method for early disease detection and monitoring.

## 2. Materials and Methods

### 2.1. Study Design

This review was conducted to evaluate the role of AI in retinal imaging for the diagnosis of cardiovascular and metabolic diseases. The methodology was guided by the Preferred Reporting Items for Systematic Reviews and Meta-Analyses (PRISMA) guidelines, which ensured a structured and transparent approach to the review process [10].

### 2.2. Indentification and Selection of Studies

A comprehensive search of the peer-reviewed literature was conducted across multiple databases, including PubMed, Scopus, and Google Scholar, to identify studies published in the last 5 years, between January 2019 and May 2024, using both manual and automated searches with MeSH terms. The search strategy was formulated using the Population, Intervention, Comparator, and Outcomes (PICO) framework (Table 1). 

Initially, research studies were hand-searched in the aforementioned databases using the following keywords: “retinal imaging”, “OCT”, “OCTA”, “fundus photography”, “cardiovascular disease”, “hypertension”, “coronary artery disease”, “metabolic disease”, and “diabetes mellitus”. The same MeSH terms were utilized, alongside their appropriate synonyms and keywords, in combination with Boolean operators (AND, OR); queries were conducted on the title, abstract, keywords, and author keyword fields, utilizing the specified keywords. The search was restricted to journal articles written in English. A methodical and systematic search strategy was employed to identify pertinent scientific papers that investigate the utilization and efficacy of retinal imaging in the diagnosis of cardiovascular and metabolic diseases, according to the PRISMA statement and checklist. The pre-established set of inclusion and exclusion criteria is given in Table 2. 

### 2.3. Data Extraction Process

The lead investigator (L.A.G.) screened the most pertinent publications based on their title, abstract, and a quick glance at the full research manuscript. Abstract screening and the literature database review were then conducted independently by two scientists. All selected articles were added to a Microsoft Excel table with columns for improved subsequent management and organization of the review: title, authors, year and journal of publication, and the specific type of publication. Articles were then structured into cardiovascular or metabolic disease identification and prognosis.

### 2.4. Overview of Studies Included in the Review

Finally, after the application of the inclusion and exclusion criteria, 37 articles (CVD: 29 articles; Metabolic Diseases: 8 articles) were chosen for the literature review, as they fulfilled all the stated requirements. The process followed for selecting the articles is summarized in the PRISMA diagram (Figure 1). 

## 3. Technological Advances in Oculomics

Ideal biomarkers are simple to determine and inexpensive, and ocular imaging has served as the foundation for an AI breakthrough due to its widespread availability and massive worldwide datasets. Color fundus and OCT imaging in ophthalmology provide a wealth of data for neural network training, and developments in multimodal imaging methods and AI have contributed to these developments. Over recent years, there has been an increasing body of data suggesting routine retinal imaging can detect early indicators of several systemic diseases (Figure 2). Thus, retinal vascular variations can aid in predicting cardiac disease and potential risk factors.

Retinal evaluation has been widely used to diagnose hypertension and diabetes mellitus, employing a semiquantitative system of evaluation. Nevertheless, over the last few decades, more accurate, non-invasive tools for assessing microcirculation have emerged. 

Retinal biomarkers, for instance ocular imaging techniques, could provide new paths for diagnosis, assessing prognosis, and managing arterial hypertension, heart failure, peripheral arterial disease, and metabolic diseases. When weighed against cardiovascular examination, retinal imaging is more straightforward, more affordable, non-invasive, and can yield high-resolution images of retinal vascularization. Furthermore, retinal vascular alterations can be an indicator of cardiotoxicity caused by medications or environmental exposure, which is a critical feature [1,2,11].

Although invasive, techniques such as fluorescence angiography facilitated the examination of retinal microvessels, while modern approaches, for example, the non-mydriatic video camera and software, permitted extensive analytical investigation of microcirculation [10]. This method and others show promise in prognostic research for cardiovascular illnesses. Further developments include the Retinal Vessel Analyzer (RVA), which investigates dynamic shifts in retinal vessel diameter [12], and scanning laser Doppler flowmetry (SLDF), which evaluates structural and functional capillary characteristics [13]. Adaptive optics retinal imaging devices, initially created for astronomy, have lately been converted for clinical application, resulting in high-resolution images of retinal microcirculation. This method can assess both retinal microvascular variations and endothelial function, but more research is needed to prove its clinical value [14].

OCT and OCTA development and improvement have been the main forces behind oculomics. The advancement of OCT technology from time domain to spectral domain and finally to swept-source has improved resolution, speed, and image quality [15]. OCTA, a readily accessible technology that was developed as an extension of OCT, which uses motion contrast to detect the flow of blood, enables the observation of depth-resolved retinal microvasculature in vivo. Volumetric data segmentation improves viewing of the deep capillary plexus and choroid and makes it easier to identify distinct retinal capillary plexuses at high quality [16]. Numerous retinal biomarkers have been established with the assistance of specific data about the anatomy and circulation of the retina produced by OCT and OCTA, as well as their link with systemic disorders [17]. Multiple imaging techniques are currently being utilized to investigate retinal blood flow patterns and microvascular network properties [18]. These methods depend on current significant advances in computer-based quantitative image analysis technologies. These advanced software programs enable extensive examinations of the morphology of the retinal vessel architecture, including microvascular width, length, quantity, and variations [18]. OCTA is particularly interesting because it provides a three-dimensional picture of retinal blood flow dynamics [19]. Recently published investigations have employed OCTA to connect modifications of retinal capillary networks to characteristics associated with cardiac remodeling [20].

New imaging techniques offer higher resolution, signal-to-noise ratio, and depth than traditional OCT, OCTA, and retinal photography methods. These methods may yield data regarding water content, proteins and lipids, cellular and redox activity, oxygen metabolism, and circulation. Underutilized imaging methods related to oculomics include AO, AO two-photon microscopy, scanning laser ophthalmoscopy, visible-light OCT, localization optoacoustic tomography, and retinal hyperspectral photography [21,22,23,24].

Semi-automated identification of retinal arterioles, venules, and the optic nerve head determines retinal microvascular characteristics. Several quantifiable retinal vascular variables can be objectively assessed: retinal caliber, arteriole-to-venule diameter ratio, tortuosity, branching angle, and fractal dimension. The software programs were shown to be extremely reliable and predictable [25].

Prior investigation in animal models of arterial hypertension has demonstrated the significance of alterations in microvascular network architecture in the onset and perpetuation of this disease [26]. The latest research findings on the frequent development of hypertension along with other cardiovascular diseases in individuals with cancer treated with anti-angiogenic medications, which influence microvascular growth and network characteristics, provide additional evidence for the critical role of retinal microcirculation in CVDs.

## 4. Ocular Biomarkers in Cardiovascular Diseases

### 4.1. Retinal Morphology Variation in Cardiovascular Disease

Since CVD accounts for about 30% of fatalities globally, it has become more important to create efficient algorithms to identify individuals who are most at risk [4]. Previous research on retinal imaging as an instrument to anticipate CVD risk has mostly focused on conventional risk variables such as age, high blood pressure, and dyslipidemia. Furthermore, researchers have looked into the use of retinal imaging to predict the risk of certain CVD subtypes, such as coronary heart disease and peripheral artery disease. This research has shown that retinal imaging can provide useful insights into CVD risk, expanding the area of risk categorization past traditional markers [7,11].

Research studies have linked retinal features, including arteriolar narrowing and venular widening, to overall mortality and CVD [27]. This suggests that changes in retinal vessels may mirror modifications in peripheral and cerebrovascular flow, shedding light on the role of small vessels in CVD onset and advancement. Tabatabaee et al. discovered that retinal arterial atherosclerosis was closely linked with the degree and extent of coronary artery disease (CAD) utilizing fundus photography [28]. Wong TY et al. discovered an association between retinal arteriolar constriction and the occurrence/development of coronary heart disease (CHD) by analyzing the diameters of each arteriole and venule on retinal images [29]. The predictive relevance of retinal arteriolar narrowing for the onset of hypertension is now thoroughly documented in multiple cohort studies conducted in different areas of the globe and among diverse races [19,30]. A meta-analysis of the majority of these investigations revealed that each decrease in retinal arteriolar diameter by 3 μm correlates with a 10 mmHg increase in blood pressure [31].

Prior research has linked vascular morphology (artery/vein diameter, branching angle, tortuosity, fractal dimension), network variations, and the presence of disease-related retinal characteristics (cotton wool sports, arteriovenous nicking, hemorrhages, microaneurysms) to CVD risk. However, focusing primarily on individual metrics may result in missing vital information and maybe underestimating the retina’s total potential to depict cardiovascular health. A broader strategy, such as artificial intelligence, could make better use of the retina’s ability as a cardiovascular risk marker.

### 4.2. Automated Idenfication and Prediction of Cardiovascular Diseases

AI can train on large datasets of retinal images to develop systems that detect and predict CVD risk factors by analyzing retinal features that correlate with cardiovascular health (Table 3). These datasets include labeled images that capture various retinal characteristics like vessel caliber, tortuosity, and the presence of microaneurysms. AI models, particularly deep learning algorithms, learn to recognize patterns and abnormalities associated with CVD. Over time, these systems can accurately predict risks by correlating retinal changes with known cardiovascular conditions, offering a non-invasive tool for early diagnosis and prevention [32].

Retinal aging was recently utilized as a substitute for CVD risk. Nusinovici et al. analyzed the retinal biological age (RetiAGE) using a DL method. They found that RetiAGE, regardless of phenotypic indicators such as impaired levels of creatinine, glucose, C-reactive protein, or chronological age, predicted the occurrence of CVDs [33]. Furthermore, another investigation studied retinal age prediction using retinal photography and discovered that the retinal age gap was strongly linked with arterial stiffness index [34] and stroke [35]. Non-invasive approaches were employed to investigate the relationship between retinal microcirculation, metabolic variables, and CVD prognosis. Previous population-based investigations discovered that narrowing of the retinal arterioles and a widening of the venules were associated with coronary artery disease, heart failure, and stroke, though the results differed by age and gender [29]. Retinal microcirculation studies at a younger age have better predictive potential, especially in females, where microvascular impairment plays a larger role in CVD [30]. The Atherosclerosis Risk in Communities Study is among the most enduring cardiovascular research efforts and has gathered 20 years of data; it concluded that retinal vascular diameter monitoring is an easy, non-invasive method that may anticipate heart failure and undesirable cardiac alterations up to almost two decades in advance, offering important information about long-term cardiovascular health [9]. Comparable outcomes across people with diabetes in the Wisconsin Epidemiologic Study of Diabetic Retinopathy supported this [36]. Nevertheless, a meta-analysis that included six studies assessing retinal vascular caliber and incident stroke for five to twelve years came to the conclusion that larger retinal venular caliber, rather than retinal arteriolar caliber, anticipated stroke [4].

Retinal fundus imaging is being investigated for stroke risk evaluation due to the similarities between retinal and cerebral microcirculations. A binary DL model with VGG19 architecture achieved great accuracy (AUC ≥ 0.966). This model received two distinct pre-processed photographs: the first one with contrast normalization and median filtering, and another from U-Net delineation for vascular imaging, with heat maps emphasizing predictive regions [37]. A separate investigation employed Inception-Resnet-v2 to estimate 10-year ischemic CVD risk, with AUCs of 0.976 and 0.876, respectively [38]. A DL system for CVD prediction was created, which includes a hybrid approach that calculates cardiac parameters such as left ventricular mass and left ventricular end-diastolic volume while also predicting myocardial infarction risk [39]. This method employs a multichannel variational autoencoder (mcVAE) and ResNet architecture, which are trained on fundus photographs and cardiac magnetic resonance datasets. The application of statistical analysis on this data yielded an AUC of 0.80, with 0.74 sensitivity and 0.71 specificity to anticipate CAD events.

In another study, researchers created a DL model known as the Deep Learning-Funduscopic Atherosclerosis Score (DL-FAS), which analyzes retinal fundus pictures to anticipate atherosclerosis. Their results demonstrated that retinal pictures could not only identify atherosclerosis but also improve traditional risk stratification values like the Framingham Risk Score when forecasting cardiovascular death. The investigation used a large-sized cohort to investigate the additional benefit of DL-FAS. Importantly, the system had a substantial link with cardiovascular mortality, even in individuals classed as intermediate risk using standard techniques [40]. This is a critical field of research, because standard risk assessments frequently underestimate the danger for individuals with asymptomatic mild atherosclerosis.

Oculomics and genomes may be used in disease diagnosis, including multimodal imaging, non-invasive biomarkers, and other examination results [41,42]. Huang et al. discovered a substantial correlation between particular retinal vascular changes and the chance of developing aneurysms, as well as the outstanding potential of retinal vascular features to predict potential aneurysm risk using a predictive, preventive, and personalized medicine method [42]. According to studies, AI can assess and detect a variety of non-ophthalmologic diseases, including chronic kidney disease, carotid artery atherosclerosis, and several CVD risk factors including blood pressure, body mass index, and hemoglobin A1c. Integrating retinal characteristics into a CVD risk algorithm increased its predicted accuracy more than integrating standard chemical indicators like high-sensitivity C-reactive protein [12].

**Table 3 biomedicines-12-02150-t003:** Summary of studies using retinal fundus imaging, optical coherence tomography (OCT), and optical coherence tomography angiography (OCTA) in subjects to identify/predict several types of cardiovascular disease. “Related” indicates that the authors did not publish an AUC value, but their results demonstrated significant connections between retinal changes and outcomes. ** The model achieved these percentages in effectively removing photos not suitable for analysis, ensuring that only high-quality images were considered for accurate evaluation. AO: adaptive optics; AUC: area under the curve; CAD: coronary artery disease; CHF: congestive heart failure; HTN: hypertension; IS: ischemic stroke; MI: myocardial infarction.

Study	Disease	Method	Dataset	Recruitment	Task	Outcome	Results	Sensitivity/Specificity
Aschauer et al., 2021 [43]	CAD	Fundus imaging, OCTA, AO	45 patients	Prospective	Identification	Clinical diagnosis	Related	-
Chang et al., 2020 [40]	CAD	Fundus imaging	15,408 images	Retrospective	Prediction	Atherosclerosis Score	AUC = 0.713	89.1%/40.4%
Cheung et al., 2021 [44]	CAD	Fundus imaging	5309 images	Retrospective	Prediction	Retinal-vessel morphological parameters	Related	-
Huang et al., 2022 [45]	CAD	Fundus imaging	145 patients	Prospective	Prediction	Vascular biomarkers and CAD-RADS	AUC = 0.739	71.1%/69.7%
Lee et al., 2023 [46]	CAD	Fundus imaging	2954 images	Retrospective	Prediction	Clinical diagnosis	AUC = 0.872	87.1%/50.8%
Matuleviciute et al., 2022 [47]	CAD	OCT/ OCTA	184 patients	Retrospective	Prediction	Retinal and choroidal thickness, retinal vascular parameters	Related	-
Nusinovici et al., 2022 [33]	CAD	Fundus imaging	129,236 images	Retrospective	Prediction	Biological age	Related	76%/55%
Ren et al., 2023 [48]	CAD	OCTA	185 patients	Retrospective	Prediction	Clinical diagnosis	AUC = 0.840/0.830	-
Son et al., 2020 [49]	CAD	Fundus imaging	44,184 images	Retrospective	Prediction	Coronary artery calcium score	AUC = 0.832	-
Wang et al., 2019 [50]	CAD	OCTA	316 patients	Prospective	Identification	Clinical diagnosis	Related	-
Cao et al., 2021 [51]	IS	OCTA	86 patients	Retrospective	Identification	Retinal-vessel morphological parameters	Related	-
Duan et al., 2022 [52]	IS	OCTA	60 patients	Retrospective	Identification	Retinal-vessel morphological parameters	Related	-
Liang et al., 2022 [53]	IS	OCTA	268 patients	Retrospective	Identification	Retinal-vessel morphological parameters	Related	-
Molero-Senosiain et al., 2022 [54]	IS	OCT/ OCTA	65 patients	Retrospective	Identification	Retinal-vessel morphological parameters	Related	-
Zhang et al., 2020 [55]	IS	OCTA	150 patients	Retrospective	Prediction	Retinal-vessel morphological parameters	Related	-
Ye et al., 2022 [56]	IS	OCT/ OCTA	66 patients	Retrospective	Identification	Retinal-vessel morphological parameters	Related	-
Arnould et al., 2020 [57]	MI	OCTA	30 patients	Prospective	Prediction	Retinal-vessel morphological parameters	Not related	89%
Diaz-Pinto et al., 2022 [39]	MI	Fundus imaging	87,476 participants	Retrospective	Prediction	Cardiac indices	AUC = 0.80	74%/71%
Sideri et al., 2023 [58]	MI	OCTA	165 patients	Prospective	Prediction	Cardiac indices	Related	-
Zhong et al., 2021 [59]	MI	OCTA	218 patients	Prospective	Prediction	Retinal-vessel morphological parameters	AUC = 0.812	65.9%/89%
Rakusiewicz et al., 2021[60]	CHF	OCTA	60 patients	Retrospective	Prediction	Clinical diagnosis	Related	-
Topaloglu et al., 2023 [61]	CHF	OCTA	50 patients	Retrospective	Prediction	Clinical diagnosis	Related	-
Zekavat et al., 2022 [62]	CHF	Fundus imaging	97,895 images	Retrospective	Prediction	Retinal-vessel morphological parameters	Related	97.4%/100% **
Chua et al., 2019 [20]	HTN	Fundus imaging/OCTA	77 patients	Prospective	Prediction	Clinical diagnosis	Related	-
Hua et al., 2019 [63]	HTN	OCTA	97 patients	Prospective	Prediction	Clinical diagnosis	Related	-
Pascual-Prieto et al., 2020 [64]	HTN	OCTA	73 patients	Prospective	Identification	Retinal-vessel morphological parameters	Related	78.3%/66.7% and 56.5%/79.2%
Peng et al., 2020 [65]	HTN	Fundus imaging /OCTA	169 patients	Prospective	Prediction	Retinal-vessel morphological parameters	Related	-
Sargues et al., 2023 [66]	HTN	OCTA	89 patients	Retrospective	Prediction	Retinal-vessel morphological parameters	Related	-
Zeng et al., 2022 [67]	HTN	OCTA	32 patients	Prospective	Identification	Retinal-vessel morphological parameters	Related	-

### 4.3. Automated Idenfication and Prediction of Cardiovascular Risk Factors

Certain metabolic diseases serve as key indicators of cardiovascular risk, playing a crucial role in assessing and preventing heart disease. Metabolic factors such as hypertension, dyslipidemia, and diabetes are well-established contributors to cardiovascular risk. Meanwhile, individual factors like age, gender, and lifestyle choices, including smoking, diet, and physical activity, also significantly influence cardiovascular outcomes. The importance of these indicators lies in their ability to identify at-risk populations early, enabling targeted interventions that can reduce the incidence and severity of cardiovascular events. When it comes to identifying cardiovascular disease, these biomarkers are vital signs [5,6,7].

Poplin et al. discovered that deep neural networks may identify various risk variables based on retinal photographs, including age, gender, smoking status, and blood pressure [66]. In Zhu et al.’s study, their model predicted age with a mean absolute error of 3.55 years, which was slightly higher than the error reported by Poplin et al. [68]. Despite this, the model showed an impressive correlation between estimated retinal age and actual chronological age, demonstrating its accuracy in age prediction [35]. In a subsequent study, Vaghefi et al. developed a convolutional neural network for predicting smoking status using just retinal pictures [69]. The studies of Betzler et al. [70] and Korot et al. [71] obtained AUC values of 0.94 and 0.93 for gender as a risk factor. Rim et al. [72] developed a DL-based biomarker predictor model that predicted 10 of the 47 systemic biomarkers tested, with particularly good results for age and sex. These findings indicate that DL can quickly extract further details from fundus images that can help in CVD risk assessment. The paper of Gerrits et al. demonstrates that DL applied to fundus photography effectively predicts important cardiovascular risk factors like age and sex, with moderate success for other cardiometabolic factors (cholesterol, LDL and HDL cholesterol, triglycerides), and, for the first time, attempts to predict glycemic markers (glucose and insulin), though performance was poor [73].

The existence and extent of diabetic retinopathy (DR) are two of the most reliable indicators of CVDs. Studies have shown substantial variation in DR (*p* < 0.00001), age (*p* < 0.002), and gender (*p* < 0.039) among individuals with high coronary artery calcium (CAC) values (≥400), which is considered to be a trustworthy prognostic tool [74]. DR detection was one of the initial proxies used to evaluate CVD using DL. It represents a risk factor and frequently manifests as complications from type 1 and type 2 DM brought on by retinal microcirculation degeneration [20]

### 4.4. Other Biomarkers in Cardiovascular Assesement

In a previous study, scientists created a RetiCAC system that used CAC as the basis for assessing CVD risk. Despite its usefulness, this research did not compare it to the commonly used QRISK3 score, which limits its relevance in healthcare decisions in the UK [75]. Following this, later research optimized RetiCAC for the UK population, culminating in the development of Reti-CVD. This technique divided participants into three risk groups and revealed substantial patterns in hazard ratios. Reti-CVD revealed a high-risk group of people who had a 10-year CVD risk of more than 10%. Furthermore, when paired with QRISK3, Reti-CVD showed greater prognostic value, especially among individuals in the borderline-QRISK3 group [76].

A significant concern is whether DL algorithms using retinal pictures can replace established cardiovascular indicators such as carotid intima-media thickness. This parameter is a proven indicator of significant cardiac events and is used to detect early atherosclerosis, particularly in intermediate-risk patients [77]. Nevertheless, its assessment is difficult, necessitating specific tools and qualified sonographers. DL systems evaluating fundus images, on the other hand, could provide a simpler approach to predicting cardiovascular risk, perhaps replacing more time-consuming techniques such as carotid intima-media thickness. A DL model was developed by Chang et al. [40] to predict atherosclerotic plaques from retinal imaging, resulting in the Deep Learning Fundus Atherosclerosis Score. This model was found to be an independent predictor of CVD deaths, even when adjusted for the Framingham Risk Score. Their investigation used the Xception framework to create a DL model that predicted atherosclerosis using retinal imaging. Originally developed on retinal imaging and carotid artery sonography data, the model had an AUC of 0.713. Importantly, the system provided additional predictive value beyond what the risk score alone could offer. Another investigation used retinal vascular biomarkers to forecast CAD, and they achieved AUCs greater than 0.692 by collecting characteristics during preliminary processing rather than straight from the retinal images [45]. This suggests that retinal imaging, combined with DL, could enhance CVD risk prediction and potentially improve early intervention strategies by integrating non-invasive retinal assessments with traditional cardiovascular risk models.

## 5. Linking Automated Prediction and Identification of Metabolic and Cardiovascular Diseases Using AI Models

Cardiovascular disease and diabetic retinopathy are closely linked, as both conditions share common risk factors, making retinal screening a valuable tool for identifying early signs of cardiovascular complications in diabetic patients. Considering the worrying growth in the prevalence of persons with diabetes mellitus and the shortage of qualified retinal specialists, an automated technique utilizing a computer-based examination of the fundus image might lessen the load on health systems in screenings for diabetic retinopathy. As a result, there is a growing interest in the development of automated analysis software that uses AI/DL to analyze fundus photography in individuals with diabetes [78] (Table 4). The IDx-DR system for automatically analyzing and proposing referral or extending screening for DR was cleared by the US FDA in 2018 as the initial AI-enabled diagnostic tool competent for independent assessment [79]. Using a DL model built on the InceptionV3 architecture, analysts were able to identify diabetic macular edema and DR, with an AUC of 0.990 on the MESSIDOR-2 dataset and 0.991 on the EyePACS dataset [32,80]. A further investigation yielded AUCs of 0.94 and 0.95 on MESSIDOR-2 and E-Ophtha datasets, respectively, for DR prognosis by combining a decision tree model with a customized convolutional neural network (CNN) for feature extraction [81]. Employing retinal imaging, a customized CNN obtained a precision of 89% in binary DR classification. Heat maps were used to emphasize significant image areas [82]. Additionally, the findings of Lee et al. [46] demonstrated that fundus images may currently be a cardiovascular risk factor in the diagnosis of CVD and that a multimodal model might perform noticeably better than a single modality model. Their multimodal networks were developed with the help of fundus images and clinical risk factors and showed improvement in the reclassification of both cases and controls [44]. Based on their findings, this strategy might be useful in predicting and preventing chronic disorders like CVDs.

Multiple large-scale studies in diabetic patients have found a correlation between alterations to the retinal microvascular morphology and the extent of both type 1 and type 2 diabetes, though the precise retinal vessel phenotype connected to diabetes is unknown. Meta-analyses have additionally discovered that retinal arteriolar diameters are independently related to body mass index (BMI), metabolic syndrome, and changes in total cholesterol, HDL and LDL cholesterol, and triglycerides [30].

A review of the Maastricht Study discovered a link between the presence of diabetes and wider retinal arteries. The lack of myogenic constriction of retinal arterioles caused by hyperglycemia is hypothesized to lead to arteriolar widening. Chronic hyperglycemia, on the other hand, might cause structural remodeling and arteriolar constriction, particularly when diabetes is paired with CVD [83]. There has been conflicting evidence in recent research on the connection between ocular variables and HbA1c levels. While AI algorithms in retinal imaging are capable of estimating HbA1c, their external reliability is limited [84]. According to OCT investigations, there is a negative association with choroidal thickness [85] and a positive correlation with macular thickness [86]. Significant correlations between HbA1c and foveal avascular zone circularity (FAZc) were discovered in OCTA investigation in diabetic individuals without retinopathy, emphasizing the early stages of microvascular damage [87]. Nonetheless, there is still no solid evidence linking OCTA parameters to long-term HbA1c management.

A recent pilot study addressed cardiovascular risk assessment in a type 2 diabetes mellitus group using DL models using fundus photography. In contrast to prior studies that employed more diverse cohorts, this one used a homogeneous database of diabetic individuals, which enabled more exact model training and a considerable increase in prediction accuracy. The scientists used a conservative prediction technique that combines retinal imaging with additional diagnostic techniques to reduce false negative ratios. This approach ensures that high-risk patients are correctly recognized, eliminating the possibility of missing those who might otherwise be misdiagnosed as low risk [30]. In the investigation of Zhang et al. [88], they developed a fundus image dataset and indicated that DL algorithms may anticipate arterial hypertension, hyperglycemia, and dyslipidemia. Additionally, their findings indicate that using DL for fundus imaging can predict blood erythrocyte characteristics such as hematocrit and mean corpuscular hemoglobin concentration. Earlier research has demonstrated that red blood cell (RBC) characteristics are connected with CVDs such as metabolic syndrome [89] and that impaired RBC characteristics can have negative impacts on retinal artery calibers [90].

In the study of Gulshan regarding retinal fundus pictures from diabetic individuals, a DL algorithm demonstrated great sensitivity and specificity in diagnosing diabetes-related retinopathy. In the first data set, the algorithm achieved 90.3% sensitivity and 98.1% specificity, while in the second data set, it reached 87.0% sensitivity and 98.5% specificity [80]. Several other DL systems have recently demonstrated great sensitivity and specificity (>90%) in diagnosing DR from retinal fundus photography, especially by utilizing high-quality images from freely accessible databases of homogeneous white populations [84,91]. Another study reported that DL models using fundus camera data could predict diabetic retinopathy, with an AUC of 0.75–0.79. For predicting poor blood glucose control, the model achieved an AUC of 0.67–0.73. These results suggest that the model is effective in identifying key cardiovascular and metabolic risk factors in diabetic patients, demonstrating its potential for early, non-invasive risk assessment [92].

**Table 4 biomedicines-12-02150-t004:** Summary of studies using retinal fundus imaging, optical coherence tomography (OCT), and optical coherence tomography angiography (OCTA) in subjects to identify/predict several types of cardiovascular disease. “Related” indicates that the authors did not publish an AUC value, but their results demonstrated significant connections between retinal changes and outcomes. AUC: area under the curve; DM: diabetes mellitus; HTN: hypertension.

Study	Disease	Method	Dataset	Recruitment	Task	Outcome	Results	Sensitivity/Specificity
Ale-Chilet et al., 2022 [93]	type I DM	OCTA	425 patients	Prospective	Identification	Clinical diagnosis	AUC = 0.58	-
Barriada et al., 2022 [32]	DM	Fundus Imaging	76 patients	Retrospective	Prediction	Coronary Artery Calcium Score	Related	91% (sensitivity)
Bernal-Morales et al., 2021 [87]	type I DM	OCT/OCTA	593 patients	Prospective	Prediction	Clinical diagnosis	Related	-
Cui et al., 2020 [94]	type II DM	Fluoroscein Angiography/OCTA/Fundus imaging	101 patients	Prospective	Identification	Clinical diagnosis	Related	96.5%/94.7%
DuPont et al., 2024 [95]	type II DM	Fluoroscein Angiography/OCTA	15 patients	Prospective	Prediction	Clinical diagnosis	Related	-
Li et al., 2020 [83]	type II DM/ prediabetes	Fundus imaging	2876 patients	Prospective	Prediction	Clinical diagnosis/risk variables	Related	-
Torabi et al., 2019 [85]	type II DM	OCT	115 patients	Prospective	Prediction	Choroidal thickness	Related	-
Zhang et al., 2020 [88]	HTN Hyperglycemia, dyslipidemia	Fundus imaging	625 patients	Prospective	Prediction	Clinical diagnosis/risk variables	AUC = 0.766; AUC = 0.880; AUC = 0.703	-

## 6. Artificial Intelligence-Driven Development of Automated Retinal Vessel Measurement Models

The development of AI-based software for retinal vessel measurement represents a breakthrough in the precise and efficient analysis of retinal vasculature. These tools utilize advanced machine learning algorithms to automatically measure retinal vessel calibers, including parameters like arteriolar and venular diameters, branching patterns, and tortuosity. Certain changes and variations in retinal vessels may be closely connected to the development of cardiovascular diseases and major cardiovascular events [96] (Table 5). By automating these processes, AI reduces the variability and potential errors associated with manual measurements, enhancing the accuracy of diagnosis and longitudinal studies. This technology also allows for large-scale screening programs, where retinal images can be quickly analyzed to identify early signs of systemic diseases like hypertension, diabetes, and atherosclerosis. Furthermore, integrating AI into retinal vessel analysis provides the potential to uncover new biomarkers [97]. This automated approach not only speeds up the analysis but also enables more consistent and reproducible results across diverse populations and clinical settings, making it a valuable tool in preventive healthcare and personalized medicine.

Tapp et al. were the first to investigate retinal microcirculation by applying fully automated software in a large sample size. They discovered that narrower retinal arteriolar diameters were substantially related to increased systolic and diastolic blood pressure. Wider venular diameters were associated with higher arterial stiffness, as evaluated by pulse wave velocity. The connections remained after accounting for additional risk variables for CVD. Retinal arteriolar tortuosity was likewise strongly associated with higher blood pressure, but venular tortuosity had a lesser relationship [98]. This shows that the structural integrity of the retinal microcirculation may indicate underlying cardiovascular stress.

The Singapore I Vessel Assessment – Deep Learning System (SIVA-DLS) is a completely automated DL model that Cheung et al. created. It achieves coefficients of correlation ranging from 0.82 to 0.95 when juxtaposed with traditional measurements, and it accurately predicts central retinal artery and vein equivalents. SIVA-DLS measures of retinal vessel diameters have been demonstrated to exceed manual measurements in their high correlation with CVD events and various risk variables [44].

## 7. Traditional Diagnostic Methods for Cardiovascular Diseases Versus AI Retinal-Based Approaches

CVDs have long been diagnosed using well-established methods such as electrocardiograms (ECGs), echocardiography, coronary angiography, and stress tests [99]. These tools have proven highly effective in identifying heart conditions, but they have limitations when compared to emerging AI-driven retinal imaging techniques. ECG and stress testing are cornerstones for diagnosing ischemic heart disease. According to a meta-analysis of over 20,000 patients from 147 studies, ECG stress testing has a 68% sensitivity and a 77% specificity in identifying CAD [100]. However, these tests are generally more effective when patients are symptomatic, which may lead to missed opportunities for early detection in asymptomatic individuals. Stress tests also come with the risk of false positives, which can result in unnecessary procedures [101,102]. In contrast, AI-based retinal imaging offers a non-invasive, more accessible approach to detecting early vascular changes linked to cardiovascular risk, even in asymptomatic individuals. AI can analyze subtle patterns in retinal vessels, potentially identifying risks earlier than other traditional diagnostic methods. Echocardiography provides detailed structural information and is crucial for diagnosing heart failure and valvular disease. The analysis of eighteen papers revealed that echocardiography had a high specificity (96%) and sensitivity (92%) for diagnosing acute heart failure [103]. A study conducted a meta-analysis of 20 studies to assess the diagnostic accuracy of brain natriuretic peptide (BNP) for heart failure, comparing it to atrial natriuretic peptide (ANP) and using echocardiographic or clinical criteria as reference standards. BNP is a highly accurate diagnostic marker for heart failure, showing greater sensitivity and specificity compared to ANP, with a pooled diagnostic odds ratio of 30.9 against clinical criteria [104]. Non-invasive coronary CT angiography [105] and calcium scoring [106] are valuable for assessing coronary artery disease and atherosclerosis with high sensitivity and specificity (80%/67% and 90%/40%) but are costly and involve radiation exposure. Despite significant advancements in AI and even with comparable specificity and sensitivity, traditional diagnostic methods continue to be the gold standard for diagnosing cardiovascular disease (Table 6).

## 8. Challenges and Future Perspectives

Oculomics is not solely limited to imaging techniques; there are various additional non-invasive biomarkers, and many others are still to be found. Tears include a few hundred different proteins and inflammatory mediators, and an analysis may be performed using minimally invasive procedures. Researchers are currently in the clinical evaluation phase and must guarantee that implementing oculomics in screening does not jeopardize safety, prejudice, or public trust. The goal is to improve outcomes, reduce medical expenses, and give patients a better quality of life. To establish confidence among patients, we must encourage standardization, investigate generalizability and algorithmic bias, and push international collaboration toward harmonized techniques. Oculomics research faces significant challenges, particularly regarding standardization and clinical utility. The lack of standardized imaging equipment and algorithms complicates data comparison, as different machines produce varying results. Moreover, translating research findings into clinical practice remains uncertain, with questions about how these findings will benefit patients and who will bear the costs [113]. This uncertainty is often referred to as the “translational valley of death”, where promising discoveries struggle to find practical application [4,30,68].

Another major concern is data sharing and privacy. Retinal images, now classified as protected health information, are difficult to share across institutions due to strict privacy regulations. This has stalled the initial excitement around using big data in oculomics. Researchers are exploring solutions, such as generative adversarial networks (GANs), which modify retinal images to protect patient identity while still allowing AI to learn from the data. Despite advancements in AI, clinicians remain crucial in designing relevant studies and interpreting results, ensuring that new technologies are effectively integrated into patient care [30,66,68].

## 9. Novel Contributions

This study shows the potential of AI-driven retinal imaging to improve the prediction and diagnosis of cardiovascular and metabolic diseases. A key contribution is the demonstration of how retinal imaging, traditionally used for ophthalmological assessments, can serve as a comprehensive and non-invasive diagnostic tool for systemic diseases. Unlike conventional methods, which often rely on invasive procedures or broad clinical metrics, AI-based retinal analysis can detect subtle microvascular changes linked to conditions like CAD, myocardial infarction, and DM. This innovative use of retinal biomarkers offers a new dimension in risk assessment, enabling earlier and more precise detection. The AI models in this study achieve high AUC values, sensitivity, and specificity, outperforming conventional risk assessment tools in certain cases.

One of the most significant innovations of this study is the potential for large-scale application in clinical settings. Retinal imaging, using techniques such as fundus photography and OCT, is already widely available and routinely used in ophthalmic exams. By integrating AI-based analysis into these imaging tools, healthcare providers can leverage existing infrastructure to screen for cardiovascular and metabolic diseases on a large scale. This approach offers a non-invasive and accessible solution, making it possible to implement widespread screening programs in primary care settings without the need for specialized equipment or invasive procedures.

The innovation of AI-driven retinal imaging lies in its ability to analyze vast amounts of data quickly and efficiently, making it suitable for large population-based screenings. With further research and standardization, the integration of retinal oculomics into routine clinical practice has the potential to revolutionize the early detection and management of systemic diseases. By addressing these gaps in current healthcare approaches, the study lays the groundwork for a transformative shift in how cardiovascular and metabolic diseases are diagnosed, making this a promising contribution to both research and clinical practice.

## 10. Conclusions

This article highlights the transformative potential of oculomics in diagnosing and managing systemic diseases through retinal biomarkers. By leveraging advanced imaging techniques like OCT and OCTA, combined with AI-driven analysis, oculomics can provide early, non-invasive detection of cardiovascular and metabolic diseases. Recent studies have demonstrated that AI models analyzing retinal images can accurately predict CAD, ischemic stroke, and myocardial infarction, with high AUC values ranging from 0.71 to 0.97 and sensitivities between 71% and 89%. Retinal imaging has also shown its capability in detecting diabetic retinopathy and predicting DM-associated complications. Advanced DL algorithms applied to fundus photography have demonstrated high sensitivity and specificity, comparable to clinical grading by ophthalmologists. The findings suggest that retinal imaging can reveal subtle changes in microvascular structure associated with conditions such as DM and CVD, offering a powerful tool for risk assessment. Moreover, AI models demonstrated high accuracy in predicting cardiovascular risk factors from retinal images. The integration of oculomics into clinical practice could revolutionize personalized medicine, allowing for earlier intervention and improved patient outcomes. However, further research is needed to standardize imaging protocols and ensure the clinical utility of these biomarkers across diverse populations.

## Figures and Tables

**Figure 1 biomedicines-12-02150-f001:**
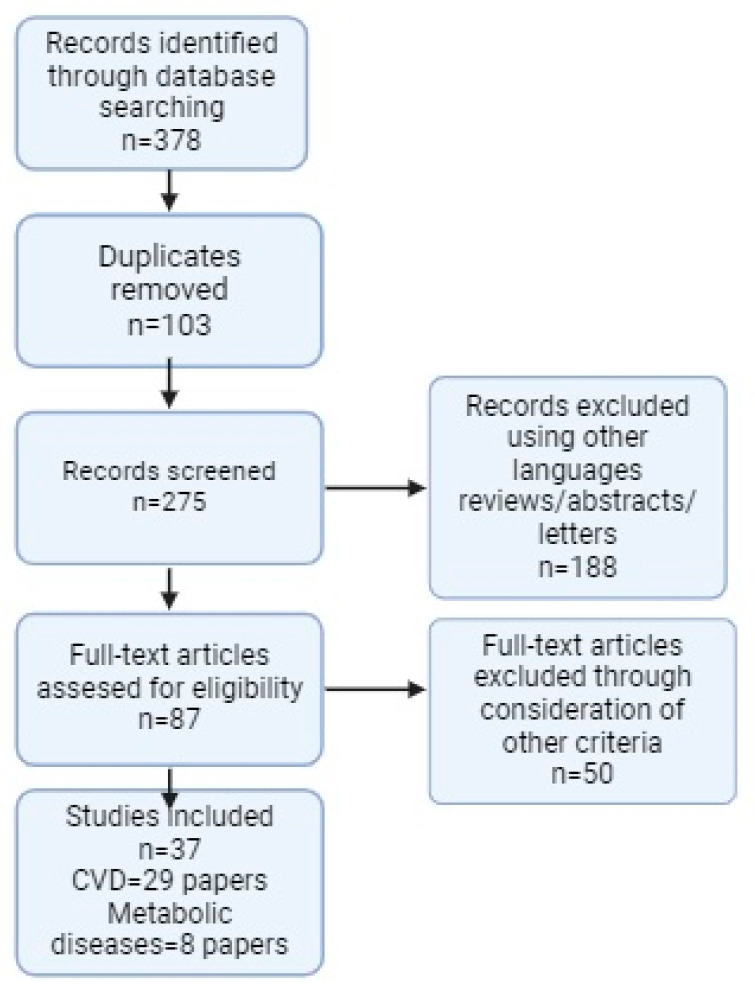
PRISMA flowchart. Created with BioRender.com.

**Figure 2 biomedicines-12-02150-f002:**
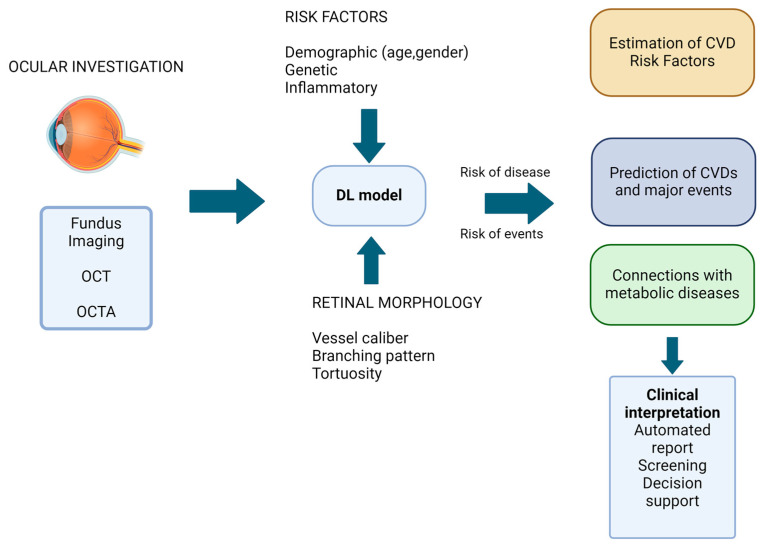
Example of methodology used in medical image analysis. Created with Biorender.

**Table 1 biomedicines-12-02150-t001:** PICO framework.

PICO	Criteria
Population	Patients at risk of cardiovascular or metabolic diseases (e.g., coronary artery disease, myocardial infarction, hypertension, diabetes)
Intervention	Retinal imaging techniques (fundus photography, OCT, OCTA) analyzed by AI and deep learning models
Comparison	Traditional diagnostic methods such as calcium score, angiography, stress test, clinical biomarkers, and ecocardiography
Outcome	High diagnostic accuracy, sensitivity, and specificity in predicting cardiovascular and metabolic diseases

**Table 2 biomedicines-12-02150-t002:** Inclusion and exclusion criteria used for this review.

Inclusion Criteria	Exclusion Criteria
Written in English	Articles in other languages
Papers utilizing retinal imaging techniques such as optical coherence tomography (OCT), optical coherence tomography angiography (OCTA), or fundus imaging	Papers with insufficient data for analysis/outcome
Papers focused on the application of AI in retinal imaging for diagnosing cardiovascular and/or metabolic diseases	Review/letters/commentaries/abstracts
Papers published after January 2018	Published before 2018
	Not relevant to the topic

**Table 5 biomedicines-12-02150-t005:** Connections between retinal characteristics and systemic diseases.

Retinal Parameter	Variation	Associated Outcome
Retinal Arteriolar Tortuosity	Increased	Linked with current blood pressure and early kidney disease
	Decreased	Associated with current blood pressure and ischemic heart disease
Retinal Venular Tortuosity	Increased	Connected to current blood pressure
Retinal Vascular Fractal Dimension	Increased	Related to acute lacunar stroke
	Decreased	Correlates with current blood pressure, lacunar and incident stroke
	Suboptimal	Associated with chronic kidney disease and coronary heart disease
Retinal Arteriolar Branching Angle	Increased	Associated with current blood pressure
Retinal Arteriolar Branching Asymmetry Ratio	Decreased	Linked to current blood pressure
Retinal Arteriolar Length-to-Diameter Ratio	Decreased	Connected to current blood pressure, hypertension, and stroke
Retinal Arteriolar Branching Coefficient	Decreased	Associated with ischemic heart disease
Retinal Arteriolar Optimal Parameter	Increased	Linked to peripheral vascular disease

**Table 6 biomedicines-12-02150-t006:** Comparison of retinal investigation methods with state-of-the-art diagnostic tools for cardiovascular and metabolic diseases.

Disease	Retinal Investigation	Comparison
CAD	Fundus Imaging, OCTA	Comparable AUC and sensitivity to current deep learning-based predictive models for CAD, such as coronary angiography and calcium scoring models with AUC > 0.8 [40,43,44,45,46,47,48,49,50,105,106].
IS	Fundus Imaging, OCTA	Moderate accuracy for ischemic stroke risk prediction compared to established methods like carotid intima-media thickness measurement and brain MRI, which have higher specificity and sensitivity. Retinal imaging offers additional predictive value but often has lower AUC [51,52,53,54,55,56,107,108].
MI	Fundus Imaging, OCTA	Some retinal imaging models show high accuracy in predicting myocardial infarction risk, comparable to traditional methods like ECG and cardiac MRI, with AUC values competitive with advanced predictive models [39,57,100,109].
CHF	Automated Retinal Vessel Measurement, OCTA	High correlation with cardiovascular events and risk factors. Shows promise as a non-invasive tool for early detection, with comparable performance to echocardiography and brain natriuretic peptide testing [60,61,62,103,104].
HTN	Fundus Imaging, OCTA	Slightly lower AUC compared to ambulatory blood pressure monitoring. However, retinal vessel analysis correlates well with cardiovascular risk factors, suggesting potential in improving predictive accuracy [20,63,64,65,66,67,110].
DM	Fundus Imaging, OCTA, Fluorescein Angiography	High sensitivity and specificity in detecting retinal complications of DM, comparable to clinical grading by ophthalmologists [32,88,93,94,111].
Hyperlipidemia	Fundus Imaging	Moderate predictive accuracy for dyslipidemia when using retinal imaging. Studies show that retinal imaging can predict lipid abnormalities but with slightly lower specificity compared to blood lipid panel tests [88,112].

## Data Availability

No new data were created.

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
