# Peer review of "Retinal Imaging-Based Oculomics: Artificial Intelligence as a Tool in the Diagnosis of Cardiovascular and Metabolic Diseases"

_biomedicines, 2024, doi:10.3390/biomedicines12092150_

Round 1

Reviewer 1 Report

Comments and Suggestions for Authors

The authors stated many things in the abstract and all are theoretical perspectives only.

There is no experimental evidence presented in the paper.

There is only qualitative data present in the abstract. No quantitative data is present in the abstract section, which is a crucial one. Even if it is a survey-type paper. 

The outcome of this article is still unclear.

It is very difficult to find the novelty. I read throughout the paper but I couldn't get any places where the author claims the originality.

There is no PRISMA studies in this article 

There are no PICO studies. 

What are the inclusion and exclusion criteria for Diagnosis of Cardiovascular and Metabolic Diseases? 

How many sample sizes are considered?

Which search string was used for conducting the Diagnosis of Cardiovascular and Metabolic Diseases?

Has the author validated the results with the state-of-the-art in Table 2 as well as Table3

Comments on the Quality of English Language

Ok

Author Response

Dear reviewer,

We thank you for your valuable comments and we will adress them point by point

1: "The authors stated many things in the abstract and all are theoretical perspectives only."

Thank you for your feedback. We understand the importance of including concrete evidence in the abstract. We have revised the abstract to better highlight the main findings of our analysis, incorporating quantitative data. However, all these studies on AI are still theoretical perspectives, with a future aim to help clinicians better diagnose CVDs and DM.

2: "There is no experimental evidence presented in the paper."

We appreciate your observation. This paper was intended as a comprehensive review of existing studies, rather than an experimental research article. Our aim was to take into consideration all novel data on incorporating AI in cardiovascular disease identification and prevention and how it may help clinicians

3: "There is only qualitative data present in the abstract. No quantitative data is present in the abstract section, which is a crucial one. Even if it is a survey-type paper."

Thank you for pointing this out. We have revised the abstract to include key quantitative data where available, such as sensitivity, specificity, and AUC

4: "The outcome of this article is still unclear."

We have revised the last part of the manuscript to clearly articulate the outcomes of the study, emphasizing the main findings and their implications for the diagnosis of cardiovascular and metabolic diseases.

5: "It is very difficult to find the novelty. I read throughout the paper but I couldn't get any places where the author claims the originality."

Thank you for your comment. We have updated the last paragraphs to better highlight the novelty of our work, particularly focusing on the comparison between AI and traditional diagnostic methods. Our aim was to provide a comprehensive synthesis that brings together various aspects of current research, offering insights that have not been aggregated in this manner before.

6."There is no PRISMA studies in this article."

We appreciate your feedback regarding the reporting standards. To our knowledge, PRISMA guidelines are typically required when conducting and reporting systematic reviews. However, we have now incorporated the PRISMA guidelines in the methodology section to provide a structured approach to our review.

7: "There are no PICO studies."

We have revised the methodology section to include a PICO framework, outlining the Population, Intervention, Comparison, and Outcome criteria used in our review to enhance the structure and clarity of the study design.

8: "What are the inclusion and exclusion criteria for Diagnosis of Cardiovascular and Metabolic Diseases?"

We apologize for any confusion. We have now included a detailed section on the inclusion and exclusion criteria used for selecting studies in our review.

9: "How many sample sizes are considered?"

We have added an entire new chapter regarding the inclusion criteria of our study. However, the tables already contained the sample sizes for each of the studies.

10: "Which search string was used for conducting the Diagnosis of Cardiovascular and Metabolic Diseases?"

We have included the specific search strings and keywords used for the literature search in the methodology section, detailing the databases searched and the inclusion of terms related to cardiovascular and metabolic diseases.

11: "Has the author validated the results with the state-of-the-art in Table 2 as well as Table 3?"

We have now expanded the discussion sections corresponding to Tables 2 and 3, providing a comparison of our findings with traditional  methods.

Once again, we thank you for your time

The authors

Reviewer 2 Report

Comments and Suggestions for Authors

The material existing in the literature is quantitively and qualitatively well presented. However, the summary part is unclear. It would be better if the sensitivity and specificity of AI in the diagnosis of CVD and metabolic diseases is determined on the basis of the retrieved material. This will allow the authors to make comparison with traditional methods and form more convincing conclusion.

Author Response

Dear Reviewer,

Thank you for your valuable feedback

 In response of your suggestions, we have added general information regarding sensitivity and specificity in the abstract to provide a clearer overview. Additionally, we have updated the tables to include sensitivity and specificity where this information was available, noting that some articles only reported the Area Under the Curve (AUC). Furthermore, we have included a small chapter comparing traditional methods with AI methods to offer a more comprehensive analysis. 

Once again, we thank you for your time. 

The authors

Round 2

Reviewer 1 Report

Comments and Suggestions for Authors

The author needs to provide quantitative information in the Conclusion section.

Still, i have not convinced the response of the author of this research question asked the 1st round "Has the author validated the results with the state-of-the-art in Table 2 as well as Table 3?"

Both of the table doesn't seem to present a brief comparison of your proposed result to the state-of-the-art.

The author has mentioned in the table of the following parameters like (Disease Method Dataset Recruitment Task Outcome Results S) but where is the comparison of proposed work?

Comments on the Quality of English Language

Ok manageble

Author Response

Dear Reviewer,

Thank you for your feedback. We have revised the manuscript to address the concerns raised:

  1. We have added quantitative data in the conclusion section to emphasize the results of retinal AI in predicting cardiovascular and metabolic diseases.
  2. We feel that tables 2 and 3 are too crowded to add a new section. However, to provide a clear comparison, we have added a new table that succinctly compares the results of retinal investigations with current state-of-the-art diagnostic tools for each disease.

We thank you once again for your time.

The authors.

Reviewer 2 Report

Comments and Suggestions for Authors

I have no more questions.

Author Response

Dear Reviewer,

Thank you so much!

Round 3

Reviewer 1 Report

Comments and Suggestions for Authors

No further changes are required. However numerous improvements in style (grammar, word spacing...) and formatting of tables and figures are required before being accepted for publication in the Journal of Biomedicine.

Comments on the Quality of English Language

ok